# Thermal imaging using sulfur polymer optics

Samuel J. Tonkin[1], Harshal D. Patel [ID][1], Jasmine M. M. Pople[1], Le Nhan Pham[1], Daniel J. Lewis[1], Batool A. Aljubran[1], Jason R. Gascooke [ID][1,2], Christopher T. Gibson [ID][3], Tilak Hewagama[4,5], Donald E. Jennings[4,6], Frank T. Ferguson [ID][4,7], Martin R. Johnston [ID][1], Witold M. Bloch[1], Alex C. Bissember [ID][8], Zhongfan Jia [ID][1], Michelle L. Coote [ID][1] & Justin M. Chalker [ID][1] ✉

Infrared thermal imaging is used in defence, security cameras, fire detection, planetary science, driver assist capabilities, medical thermography, and other safety applications. Unfortunately, the lenses for infrared cameras are made from expensive or restricted materials such as germanium, silicon, or chalcogenide glass. Furthermore, these inorganic lenses are made by low throughput milling processes, and they are difficult to repair or recycle. There is a need for low cost and sustainable lens materials that can be mass-produced to prescription. Sulfur-derived polymers, made from widely available elemental sulfur, are promising candidates due to their high refractive index and mid-wave infrared (MWIR) and long-wave infrared (LWIR) transparency. However, most of these polymers reported to date are still limited in their LWIR transmittance and the glass transition temperature required for shape persistence. Recently, a polymer containing a sulfurized norbornane microstructure was predicted by Pyun, based on theoretical considerations, to address these issues. However, this polymer has not yet been made due to complex side reactions encountered in previously attempted syntheses. Here, we overcome these challenges and prepare this polymer for the first time, demonstrate methods for high throughput molding and recycling, and validate its use as a lens in a long-wave thermal imaging camera.

Thermal imaging is used in security cameras[1,2], fire detection[3], Earth system and planetary science remote sensing[4], driver assist features in automobiles[5], medical thermography[6], and other safety applications[1,2]. Unfortunately, lenses for thermal cameras are made from expensive or restricted materials such as germanium, silicon, or chalcogenide glass, which limits broader use[7]. There is a need for low cost and sustainable alternatives that can be rapidly mass-produced to prescription. Sulfur-rich polymers, made from widely available elemental sulfur, are promising candidates due to their high

refractive index and mid-wave infrared (MWIR, 3–5 μm) and long-wave infrared (LWIR, 7–14 μm) transmittance[8–15]. To this end, pioneering studies by Pyun, Norwood, and others have shown sulfur-rich copolymers are useful as IR transparent windows and lenses[9,13], photonic crystals[16], waveguides[17,18], polarizers[19], Bragg gratings[20], and other integrated photonic devices[17]. In addition to their optical properties, these polymers are attractive because they are derived from low-cost elemental sulfur, and they can be molded and repaired due to dynamic S–S bonds in their structure[21]. In contrast,

[1]College of Science and Engineering, Flinders University, Adelaide, SA, Australia. [2]Australian National Fabrication Facility, South Australia Node, College of Science and Engineering, Flinders University, Adelaide, SA, Australia. [3]Adelaide Microscopy, The University of Adelaide, Adelaide, SA, Australia. [4]NASA Goddard Space Flight Center, Greenbelt, MD, USA. [5]Department of Astronomy, University of Maryland, College Park, MD, USA. [6]Science Systems and Applications, Inc. (SSAI), Lanham, MD, USA. [7]Catholic University of America, Washington, DC, USA. [8]School of Natural Sciences—Chemistry, University of Tasmania, Hobart, TAS, Australia. ✉e-mail: justin.chalker@flinders.edu.au

germanium, ZnSe, or silicon lenses are made by slow milling processes and they are not easy to repair.

Despite these advances, there is still a need for sulfur-rich polymers with high glass transition temperatures ($T_g > 100\,°C$)—to impart shape persistence to the lens—and much higher transparency to 7–14 µm light. The increased LWIR transparencies are especially important if these materials are to compete with commercial cameras and sensors or light collection apertures in drones and SmallSat instruments. Recently, a copolymer containing a sulfurized norbornane core (1) was predicted by Pyun, based on theoretical considerations, to address these needs[13,22]. However, this polymer has not been made due to complex chemistry encountered in previously attempted syntheses[13,23]. Here, we overcome these challenges and synthesize this polymer for the first time and validate its use in thermal imaging.

## Results and discussion

In 2019, Pyun and co-workers proposed polymer 1 as a candidate for infrared optics applications (Fig. 1a)[13]. The symmetry of the rigid norbornane core and high sulfur content were predicted to impart high MWIR and LWIR transparency, based on calculated infrared spectra of structurally related model compounds[13,22]. In principle, polymer 1 could be made by the reaction of sulfur and norbornadiene. However, this reaction is challenging due to the volatility of norbornadiene and premature vitrification under solvent-free conditions (Supplementary

**Fig. 1 | Rearrangements in the attempted copolymerization of sulfur and norbornadiene. a** Polymer **1** was predicted by Pyun and co-workers, on theoretical grounds, to have infrared transparencies suitable for thermal imaging applications. **b** The reaction between norbornadiene and sulfur does not provide **1**. Several complex rearrangements occur that are detrimental to LWIR transparency, especially the formation of cyclopropane units that strongly absorb at 804 cm⁻¹. All of these intermediates can undergo homopolymerization or copolymerization with sulfur. X-ray crystal structures are shown for purified, isolated intermediates of the reaction of sulfur and norbornadiene. **c** The copolymerization of monomer **9** and sulfur is shown as a representative route to **1**. This ring-opening polymerization involves only S–S exchange reactions, as all C–S bonds are preformed with defined stereochemistry in the monomer synthesis. This reaction provides the first synthetic access to **1**.

Figs. S1 and S2)[13]. We therefore explored solution-phase polymerization of sulfur and norbornadiene (Supplementary Fig. S3), but found this approach was plagued by complex rearrangements (Fig. 1b)[24]. In particular, the formation of cyclopropane units resulted in a strong absorbance at 804 cm⁻¹ that drastically reduced LWIR transparency[25,26]. As shown in Fig. 1b, several of these intermediates were detected and isolated through a combination of meticulous flash column chromatography, selective precipitation and trituration, and recrystallizations. X-ray crystallography confirmed the structures of several of these intermediates unambiguously for the first time (Fig. 1b and Supplementary Figs. S26–S29). Simulated infrared spectra of **2**–**9** were computed using density functional theory (DFT) to gain insight into the key absorbances that might also be included in polymers made from these monomers (Supplementary Fig. S75). Importantly, we demonstrated that each of these intermediates can undergo homopolymerization or copolymerization with sulfur (Supplementary Fig. A, page S4). Therefore, any polymer derived from the reaction of norbornadiene and sulfur will contain complex microstructures, including cyclopropane units with a strong absorbance at 804 cm⁻¹.

These rearrangements are consistent with previous studies by Bartlett and Ghosh[24], and Pyun also reported cyclopropane microstructures in copolymers made from sulfur and other norbornadiene derivatives[25,26]. Recently, He, Xia, and associates reported an intriguing mechanochemical reaction of sulfur and norbornadiene, reportedly resulting in the synthesis of **1**[23]. However, the product had far lower LWIR transparency than expected[23]. We carried out a control reaction between sulfur and norbornadiene in a planetary ball mill to see if rearrangements also occurred in the mechanochemical approach. Indeed, cyclopropane rearrangement products formed, as supported by infrared spectroscopy and GC-MS analysis after reducing the product with LiAlH₄ (Supplementary Figs. S33–S38). In fact, the same product distributions were observed for mechanochemical and solution phase reactions, indicating polymer **1** is not formed by either method (Supplementary Fig. B, page S5).

The complex reaction of sulfur and norbornadiene motivated Pyun and others to develop structurally related monomers that do not suffer the same rearrangements (Supplementary Fig. C, page S6)[13,26,27]. While these polymers possess useful $T_g$ values > 100 °C and showed promise in thermal imaging[13,26,28], their LWIR transparencies are still relatively low. We, therefore, wanted to synthesize **1** to compare its performance in LWIR imaging, as **1** has fewer C–H and C–C bonds that otherwise contribute to LWIR absorbance. At the same time, **1**—with its rigid norbornane microstructure—was anticipated to exhibit a high $T_g$ required for shape persistence of lenses and thermal stability. To access **1**, our approach was to use monomers such as **7**–**10** that contain pre-installed trisulfide or pentasulfide rings. These compounds do not form cyclopropane units when copolymerized with sulfur (Supplementary Fig. A, page S4). For subsequent studies of these polymers in infrared optics, we focused on monomers **9** (Fig. 1c) and **10** (Fig. 2). These monomers have all C–S bonds pre-installed with defined stereochemistry. Furthermore, these monomers are soluble in molten sulfur, which facilitates homogeneous and solvent-free ring-opening copolymerization through S–S exchange reactions.

To synthesize **9** and **10**, norbornadiene and sulfur were reacted in a 1:1 DMF:toluene solvent system at 120 °C in the presence of a [Ni(NH₃)₆]Cl₂ catalyst (Fig. 2a)[29–31]. After 24 h of reaction, polymer byproducts were precipitated in water and soluble organic products were extracted into hexane. Conveniently, **9** and **10** precipitated from the organic solvent overnight. The overall yield was 10%, but the low cost of the reagents and operational simplicity of the method allowed access to decagram quantities of **9** and **10**. The monomers in the mixture could be distinguished by nuclear magnetic resonance (NMR) spectroscopy and differential scanning calorimetry (DSC) (Fig. 2b–d). The ratio of compound **9**:**10** was typically about 5:1 in this synthesis. While pure crystals of **9** could be prepared by recrystallization from

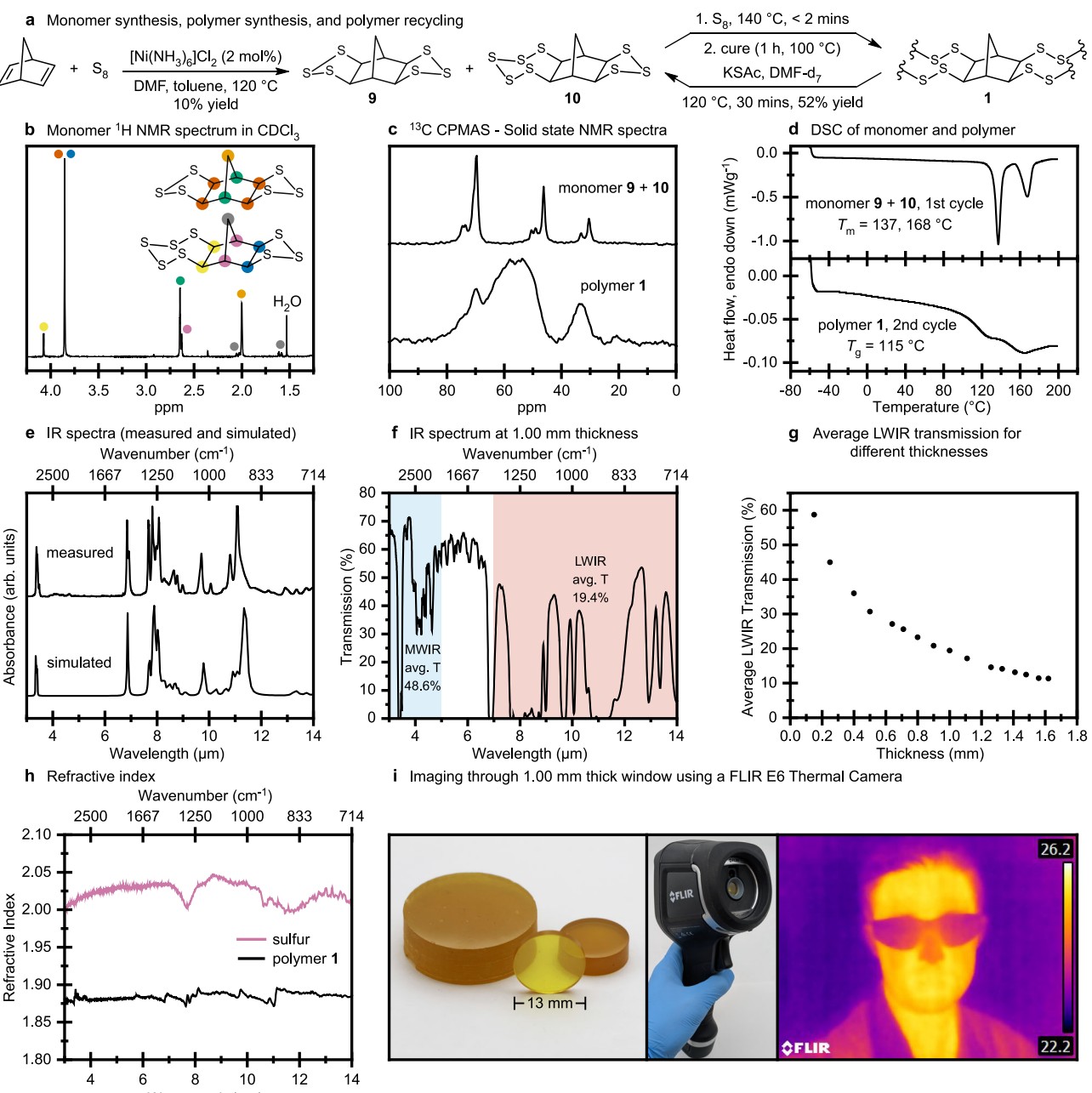

**Fig. 2 | Synthesis and characterization of polymer 1. a** Synthesis of monomers **9** and **10** and their copolymerization with elemental sulfur to form polymer **1** (81% sulfur by mass). Recycling by catalytic depolymerization was also achieved. All characterization data provided in this figure are for the polymer **1** composition of 81% sulfur. **b** ¹H NMR spectrum of monomers **9** and **10** in deuterated chloroform. **c** Solid state ¹³C CPMAS NMR spectra of monomers **9** and **10** and polymer **1**. **d** DSC thermogram of monomers **9** and **10** and polymer **1** (81% sulfur). **e** Measured and simulated infrared spectra for polymer **1**. **f** FTIR spectrum through a 1 mm thick window polymer **1** (81% sulfur). The average transmittance over the MWIR (3–5 μm) and LWIR (7–14 μm) regions are shown. **g** Average LWIR transmittance for a polymer **1** window of varying thickness. **h** Refractive index of elemental sulfur and polymer **1** (81% sulfur), measured by reflectance spectroscopy. **i** Polymer **1** was cast into molds to form windows and preforms. LWIR imaging through a 1.0 mm thick window of **1** using a FLIR E6 camera.

hexane, it was found that a mixture of **9** and **10** was suitable for polymerization. The sulfur content in polymer **1** was controlled by considering the total sulfur content in the monomer mixture, and the amount of elemental sulfur used in the polymerization.

In the polymerization, sulfur was first melted at 140 °C and degassed in a vacuum oven. Monomers **9** and **10** were added to the sulfur and dissolved. Due to the highly crosslinked network formed in the polymerization, the working time was very short, with vitrification observed within 3 min at 140 °C. To cast polymer **1**, it was stirred manually at 140 °C for 45 s after the addition of **9** and **10** to the molten sulfur. The reaction was then heated for an additional 15 s before

pouring into a mold and transferring to an oven where it was cured at 100 °C for one hour. While a range of compositions were accessible, 81% sulfur by mass provided the optimal combination of thermomechanical and optical properties (Supplementary Fig. S58). This composition balances the high $T_g$ imparted by the norbornane core, while having sufficient sulfur for a higher refractive index and LWIR transmission. Polymer **1** with 81% sulfur by mass was used in subsequent experiments for these reasons.

The cured polymer product was characterized by solid state NMR spectroscopy, DSC, and FTIR spectroscopy (Fig. 2c–f). Comparison of the solid state ¹³C-NMR spectrum of **9** and **10** to the product revealed

broadened peaks, which is expected in a polymer (Fig. 2c). The characteristic peaks of cyclopropane ($\delta = 10–15$ ppm) were absent, indicating rearrangement did not occur. DSC analysis of **1** did not reveal any melting transitions, indicating complete consumption of elemental sulfur and monomers **9** and **10** (Fig. 2d). Despite a high sulfur content, polymer **1** had a high glass transition temperature of 115 °C when measured by DSC and 154 °C when measured by dynamic mechanical thermal analysis (DMTA) (Supplementary Fig. S62), which is important for shape persistence and thermal stability in precision optics.

Simulated infrared spectra of **1** and oligomers of **1** (monomer to hexamers) were obtained using density functional theory (DFT, Supplementary Fig. S71), while longer polymers (20-mers and 30-mers) were studied using GFN2-xTB (Supplementary Fig. S74). Convergence of the key spectral features with respect to chain length was observed by the hexamer stage, although minor differences in peak intensities—particularly in the C–H stretching region—persisted across all chain lengths due to variations in the proportion of mid-chain versus end-chain units. No significant differences were observed in the simulated spectra of linear versus cross-linked models of **1** at the hexamer stage (Supplementary Fig. S71). Gratifyingly, FTIR spectroscopic analysis of windows of polymer **1** were nearly identical to these calculated spectra of polymer **1** (Fig. 2e). Importantly, the absorptions of cyclopropane units at 804 cm$^{-1}$ and 3065 cm$^{-1}$ were not present.

In addition to making polymer **1**, a depolymerization method was developed for chemical recycling of offcuts or damaged windows or lenses (Fig. 2a). Potassium thioacetate was used as a nucleophilic catalyst to break S–S bonds in **1** and reform monomers **9** and **10** through S–S exchange reactions (Fig. 2a). A complementary thermal depolymerization was also developed in solution (Supplementary Fig. S66). In both cases, polymer **1** is initially insoluble, but as the monomers reform, they dissolve into solution. Dilute conditions were used to favor depolymerization and slow any unwanted reactions of the regenerated monomers. These depolymerizations are an important capability for lifecycle management of synthetic polymers[32].

With ample amounts of monomers **9** and **10** in hand, polymer **1** was fashioned into freestanding windows and preforms by casting the polymerization mixture into suitable molds and curing. Due to the high glass transition temperature of **1**, it could be polished to provide optical quality surfaces using a micromesh polishing kit and micro gloss polish (Supplementary Fig. S69 and page S83). FTIR spectroscopic analysis of a 1.00 mm thick window of **1** prepared in this way revealed excellent transmission in both the MWIR (3–5 μm) and LWIR (7–14 μm) regions with an average transmission of 48.6% and 19.4%, respectively (Fig. 2f). Additional transmission was measured across a range of thicknesses (Fig. 2g), with windows of **1** possessing more than 11% LWIR transmission at thicknesses as great as 1.6 mm and more than 58% LWIR transmission for 0.15 mm thin windows. The LWIR transmission levels are the highest reported to date for polysulfide polymers having a $T_g > 100$ °C (Supplementary Fig. D and Supplementary Table S8, pages S7 and S122)[8,12–15,26,27,33]. The high sulfur content of polymer **1** also imparted an average refractive index of $n = 1.87$ over the MWIR and LWIR regions (Fig. 2h). As a qualitative demonstration of the potential of **1** in infrared imaging, a 13 mm diameter, 1.00 mm thick window of **1** was mounted on the front of a FLIR E6 thermal camera. The excellent LWIR transmission of **1** allowed high quality imaging of low temperature objects or human subjects through the polymer window (Fig. 2i). In this demonstration, the thermal camera is still operating with a germanium lens, so it merely demonstrates LWIR transparency of the polymer **1** window. Therefore, the next goal was to make lenses from polymer **1** and test them as replacements for germanium or silicon in an LWIR thermal imaging camera.

Accordingly, a range of lenses were made from polymer **1** for evaluation as replacements for the silicon lens in the FLIR Lepton 3.5 thermal imaging module. All lenses were plano convex to simplify molding, with a maximum thickness of 0.8 mm at their center. Seven lenses were evaluated, with focal lengths ranging from 1.5 to 5.0 mm and f numbers ranging from 0.75 to 2 (Supplementary Fig. S76). These lenses provide horizontal fields of view from 65° to 22° when mounted on the imaging module. Commercially available glass lenses with these specifications were used as positives for preparing silicone molds used to cast polymer **1** (Supplementary Fig. S77). The resulting lenses were used directly without polishing and no antireflective coating was applied. The silicon doublet lens was removed from the FLIR Lepton 3.5, and the polymer **1** lens was mounted using a custom-made 3D printed holder (Fig. 3a and Supplementary Fig. S79).

To test focus and resolution in accordance with the standard procedures recommended by Pyun[28], a USAF-1951 target was laser cut from a 3 mm thick acrylic substrate. The mask was placed 5 cm from a hotplate and the thermal camera was placed 20 cm from the mask for all lenses except lens 7, which was placed 30 cm from the mask. Thermal sensitivity was investigated by taking images of the target for varying hotplate temperatures (Fig. 3b and Supplementary Figs. S91–S97). All lenses showed good resolution and sensitivity even at temperatures as low as 40 °C. The excellent transmission of the polymer lenses made it possible to image human subjects and low temperature objects (Supplementary Fig. S103).

Further quantitative assessment of polymer **1** lenses was carried out to determine the noise equivalent thermal difference (NETD), which is a measure of thermal sensitivity in the imaging system. For polymer **1** lenses with f-numbers of 0.75 and 0.8, the NETD in the FLIR Lepton 3.5 camera was 62.1 mK and 63.0 mK, respectively. This is comparable to the commercial silicon doublet lens (f/1.1) designed for the FLIR module, which exhibited a measured NETD of 53.6 mK when tested under the same conditions (Supplementary Table S7 and Fig. S99). Another quantitative measurement of the polymer **1** lenses involved relative illumination testing (Supplementary Figs. S100–S102). This metric indicates how the intensity of light changes from the center to the edge of an image. Because all polymer **1** lenses used in this study are simple plano convex designs, the lenses with a wide field of view exhibited a decrease in illumination at the edges of the image. With that said, polymer **1** lenses with a longer focal length performed much more similarly to the optimized silicon doublet lens. These results are promising in that more sophisticated lens design could further improve imaging with polymer **1** optics, including freeform optical designs.

The long-term stability of polymer **1** and polymer **1** optics are critical for consistent production of high-quality thermal images. Therefore, we measured the $T_g$, TGA profile, and refractive index for samples of polymer after 6 or more months of storage. Negligible differences were observed between these samples and freshly prepared samples of polymer **1** (Supplementary Figs. S109–S111). Additionally, a 12-month-old polymer lens was re-tested in the FLIR Lepton 3.5 module, and the image quality was the same as when first tested 1 year prior (Supplementary Fig. S112). We attribute the stability of **1** to its high $T_g$, which is consistent with studies of inverse vulcanization that suggest high $T_g$ is correlated with greater thermal stability and reduced sulfur loss during processing and storage[34].

The only prior demonstrations of freestanding polysulfide lenses serving as the sole refractive optical element in a LWIR camera were reported for polymers **15** and **17** (Supplementary Fig. C, page S6)[26,28]. While these advances were critical in realizing the promise of sulfur polymer optics, the LWIR transparency of these polymers limited the image quality for low temperature targets. In contrast, polymer **1** with its higher LWIR transparency, could afford excellent resolution and contrast at ambient temperatures, providing high quality still images and even video (see Supplementary Movie 1). Therefore, LWIR imaging using a polymer **1** lens is a new benchmark for this class of materials[8] and validates Pyun's original hypothesis of the potential utility of polymer **1**[13]. An expanded comparative assessment of polymer **1** and

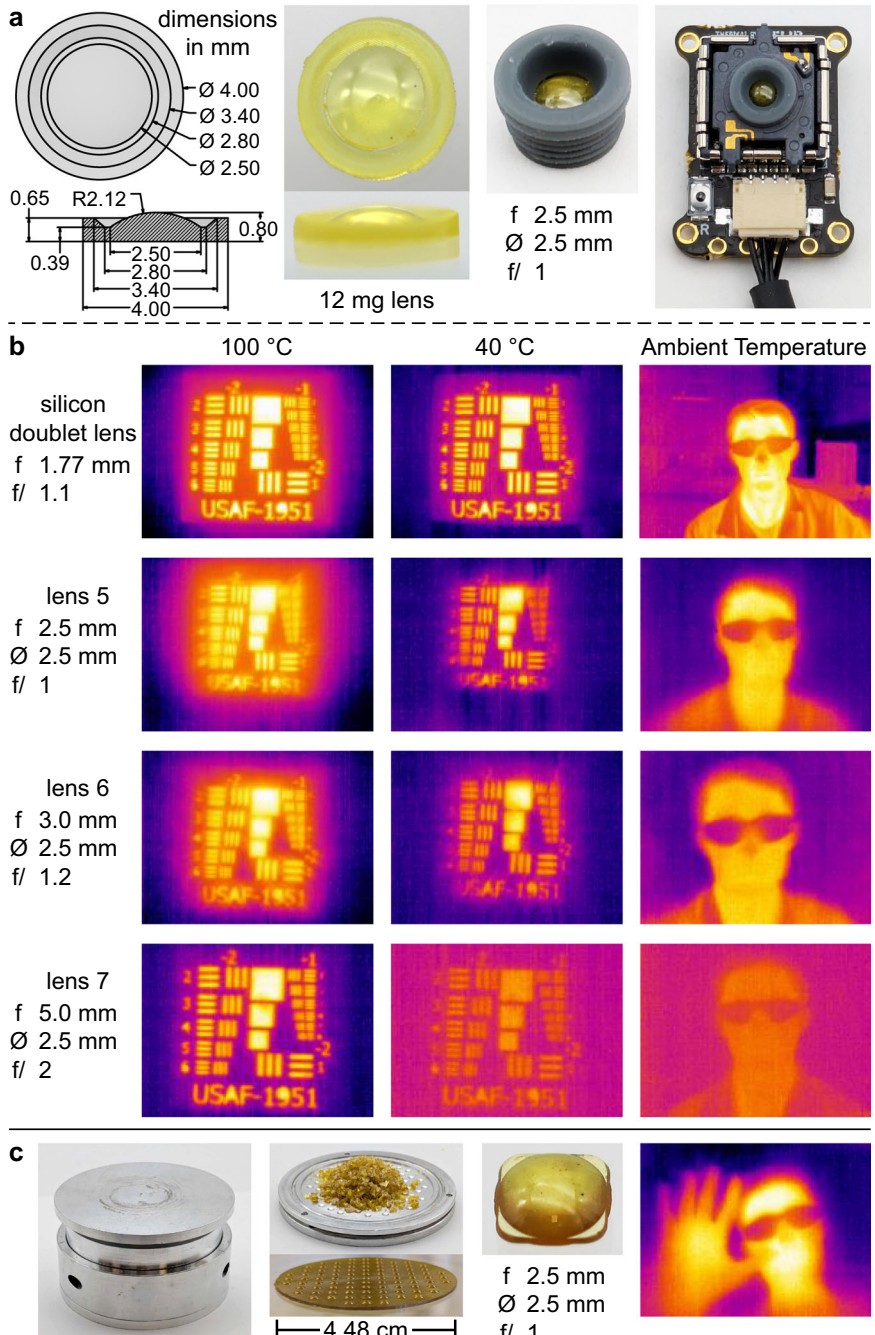

**Fig. 3 | LWIR thermal imaging with polymer 1 optics. a** Schematic and images of a polymer **1** lens (81% sulfur), mounted on the FLIR Lepton 3.5 using a 3D printed holder. **b** Imaging using polymer **1** lenses (81% sulfur) on the FLIR Lepton 3.5 (3 separate lens designs shown) and a comparison to the commercial silicon doublet lens optimized for the FLIR Lepton 3.5 LWIR imaging module. The camera with the silicon doublet was placed 12 cm from mask, lens 5 and 6 were placed 20 cm from mask and lens 7 was placed 30 cm from mask. **c** Reactive compression molding of polymer **1** to prepare a monolithic array of 89 lenses. A single lens cut from the array is shown as well as the IR image taken using this lens after mounting on the FLIR Lepton 3.5 thermal imaging module. Additional resolution and sensitivity analysis is provided in the Supplementary Information.

previously reported sulfur polymer optics is provided in the Supplementary Information (Supplementary Table S8).

The lenses in Fig. 3a were made one at a time using a casting process. For higher throughput manufacture, we envisioned molding pre-made polymer **1** into many lenses at once by reactive compression molding[35]. It was hypothesized that with an optimal temperature and pressure, particles of **1** could be consolidated through thermally-induced S−S metathesis and pressure to form the lens. A custom aluminum die was designed and machined so that 89 plano convex lenses (2.5 mm focal length, f = 1) could be made in a single hot-pressing operation. Next, particles of pre-made polymer **1** (2.5 g, ~1 mm particle size) were added to the mold. The die was then heated to 185 °C and equilibrated for 1 min before compressing at 20 MPa for 2 min. Remarkably, this short cycle time is sufficient for the polymer **1** particles to consolidate cleanly, with no visible scarring. The resulting single polymer piece could be easily removed after cooling and the lenses were separated (Fig. 3c). The imaging performance of these lenses was comparable to the lenses made by casting in a silicone mold (Fig. 3c and Supplementary Figs. S104−S108). The ability to mold pre-made polymer **1** bodes well

for high-throughput lens manufacture, which could be an advantage in some scenarios in comparison to slower, subtractive milling processes typically used for inorganic optics. Reactive compression molding can also be used to recycle off-cuts or damaged lenses, which is an advantage over germanium and silicon.

In summary, polymer **1** was synthesized and characterized for the first time, allowing experimental validation of its utility in infrared imaging and video. The material has excellent LWIR and MWIR transparency and a high glass transition temperature. Lenses made from **1** were effective in LWIR imaging of subjects at room temperature, which is an important advance in sulfur polymer optics[8], and a requirement for the many thermal imaging applications that detect or image humans or wildlife; such optical elements are also relevant in remote sensing applications in Earth system and lunar and planetary surface studies.

Future work will focus on upscaling the synthesis of **1** and making more complex optics, to deliver increased image quality required for higher-end applications in navigation, defense, security systems, space applications, firefighting, and high-temperature monitoring in furnaces and nuclear reactors. For preparing more complex prototype lenses, the high $T_g$ and stability of **1** is anticipated to allow construction of more complex architectures by diamond point turning of polymer preforms. The integration of antireflective coatings will also be examined to further improve LWIR transmission. One approach we will explore is varying the amount of sulfur in the polymer at the surface of the lens, with an aim to make a gradient refractive index at the boundary of a monolithic lens, rather than applying a traditional antireflective coating. This approach is anticipated to overcome challenges of peeling that can occur with conventional coatings. Further investigations into design of the organic component of these polymers are also underway as part of efforts to further reduce LWIR absorbance. The broader goal is to provide low-cost and sustainable alternatives to germanium, silicon and ZnSe for thermal imaging applications.

## Methods

### Synthesis of monomers 9 and 10
A 250 mL round-bottom flask was charged with sulfur (9.45 g, 36.88 mmol $S_8$), norbornadiene (5.00 mL, 4.53 g, 49.15 mmol) and [Ni(NH$_3$)$_6$]Cl$_2$ (228 mg, 0.984 mmol, 2 mol%). 50 mL of DMF and 50 mL of toluene were added along with a magnetic stirrer. The flask was equipped with a condenser and heated at 120 °C for 24 h. After this time, the solution was poured into a beaker containing 250 mL of hexane and 250 mL of distilled water. Polymeric material precipitated and settled in the aqueous layer. The organic layer was then filtered through celite. The orange-colored filtrate was collected and washed three times with 250 mL of distilled water to remove any remaining DMF. The organic layer was then dried with magnesium sulfate and filtered into a round-bottom flask. The washing and filtering steps were done quickly because **9** starts to precipitate during the workup procedure. The round-bottom flask was then sealed and left overnight. Monomer **9** would precipitate as a yellow powder over 24 h and would usually adhere to the side of the glass. The hexane solution was carefully decanted to avoid any loss of monomer **9**. Toluene was then added to the round-bottom flask to aid in the removal of **9**. Sonication was required to remove it from the glassware. Toluene was then removed by rotary evaporation. ¹H NMR analysis indicated that **9** co-precipitated with molecule **10** (Supplementary Fig. S25). If a pure crystalline sample was required, a hot recrystallization in hexane gave pure **9** with a melting point of 179 °C. A yield of ~10% of **9** and **10** was obtained using this protocol. NMR spectra and X-ray structures of **9** are provided in Supplementary Fig. S23 and S28. The X-ray diffraction data was obtained on the MX136 and MX237 beamlines at the Australian Synchrotron, Victoria, Australia[36,37].

### Polymer 1 synthesis
Sulfur (300 mg, 9.36 mmol S atoms) was added to a 5 mL vial. The vial was added to a preheated oil bath at 140 °C. The sulfur was heated for 3 min at 140 °C, causing it to melt. The vial of molten sulfur was then transferred to a vacuum oven preheated to 140 °C. The pressure in the oven was decreased to approximately 2 mbar and held for approximately 30 min with continued heating at 140 °C. After degassing the sulfur, the sample was returned to atmospheric pressure and immediately returned to the 140 °C oil bath. Monomer **9** (445 mg, 1.56 mmol) was then added to the molten sulfur in one portion. This quantity of sulfur and **9** provided a final polymer product containing 81% sulfur by mass. The reaction mixture was stirred gently with a heated spatula to ensure that **9** was fully mixed with the molten sulfur. This manual stirring was continued for a total of 45 s from the time monomer **9** was added. After this time, the reaction was homogenous and not mixed further. After 60 s of total reaction time, the reaction mixture was poured into a silicone mold that was preheated to 140 °C. The mold was then placed in a 100 °C oven where it was left for 1 h to cure. If a mixture of monomer **9** and **10** was used in the copolymerization with sulfur, ¹H NMR spectroscopy was first used to quantify the ratio of **9** and **10**, and the amount of elemental sulfur was adjusted accordingly to provide the desired composition in the final polymer.

### Polymer 1 windows
Silicone molds with a 13 mm diameter opening were prepared by pouring M4504 silicone into a 3D printed part of the desired dimensions. All 3D printed parts were designed on Autodesk Inventor and printed on a Phrozen Sonic Mini 8 K. To cast polymer **1** windows, the silicone mold was placed in a preheated oven (140 °C) and polymer **1** was prepared by the method described above. The liquid prepolymer was poured into the mold and immediately placed in an oven to cure at 100 °C for 1 h. After curing, the window was removed from the mold and polished on both sides using micromesh sandpaper ranging from 1500 to 12,000 grit to give an optical quality finish (Fig. 2i and Supplementary Fig. S40).

### Polymer 1 lenses
To cast lenses from polymer **1**, molds were first prepared using glass positives. 3D printed parts were used to hold the positives and allowed for the incorporation of carrier material to the lenses. The carrier material provided mechanical strength to the lens and aided in molding and mounting of the lenses. The 3D printed part also allowed for the incorporation of a funnel and air outlet which helped with consistency when casting polymer **1**. The final mold consisted of two silicone parts that were held together with a clamp when casting. Additional details and images of the molds are provided in Supplementary Fig. S77. To cast polymer **1** lenses, the silicone mold was preheated to 140 °C in an oven and polymer **1** was prepared by the method described above. The liquid prepolymer was poured into the mold and immediately placed in an oven to cure at 100 °C for 1 h. After curing, the funnel and outlet were removed by scoring a line with a scalpel and breaking off excess polymer **1**. Any excess polymer was removed using a hand file until only the lens and carrier material remained.

### Supplementary methods
Additional methods, characterization and imaging is provided as Supplementary Information. The Supplementary Information document includes full experimental details, mechanistic studies, monomer and polymer synthesis and characterization, theoretical methods and computational studies for simulated infrared spectra, and additional optical characterization. Supplementary Movie 1 illustrates the use of lens 6 and the FLIR Lepton 3.5 in capturing video of the operation of a kitchen appliance.

## Data availability

The data generated in this study are provided in the Supplementary Information and in Source Data files. This includes source data for Figs. 1–3: cif files for X-ray structures, NMR and IR spectra, DSC thermograms, and STEP files for 3-D printed and machined parts. Matlab scripts used for NETD testing and relative illumination testing are also provided. Crystallographic data for compounds **2**, **7**, **8**, and **9** have been deposited in the Cambridge Structural Database (CCDC numbers 2305219, 2289708, 2305218, and 2289709). All data are available from the corresponding author upon request. Source data are provided with this paper.

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

## Acknowledgements

The authors acknowledge financial support from the Flinders University High Impact Collaborative Research Development Fund, the Australian Research Council [DP200100090 (J.M.C.), DP210100025 (A.C.B., M.L.C.), DP230100587 (J.M.C, Z.J.), FT200100049 (A.C.B.), FT220100054 (J.M.C.), FT240100330 (W.M.B) and CE230100021 (M.L.C.)], and the Australian Department of Education [AE240300053 (J.M.C., S.J.T., H.D.P.)]. Parts of this research were undertaken on the MX136 and MX237 beamlines at the Australian Synchrotron, Victoria, Australia. The authors also acknowledge generous supercomputing time on Flinders Deepthought and the National Facility of the National Computational Infrastructure, and the equipment and technical expertise provided by Microscopy Australia, Flinders Microscopy and Microanalysis, the Australian National Fabrication Facility and Flinders Engineering Services.

## Author contributions

Monomer synthesis and mechanistic investigations were led by S.J.T. and H.D.P, with support from J.M.M.P., D.J.L., and B.A. Polymer synthesis was carried out by S.J.T. and H.D.P. Spectroscopic characterization was done by S.J.T., H.D.P., J.R.G., C.T.G., T.H., D.E.J., F.T.F and M.R.J. X-ray analysis was completed by W.M.B. Mechanochemical studies were completed by H.D.P. and A.C.B. Simulated infrared spectra were calculated by L.N.P. and M.L.C. Polymer recycling was done by H.D.P and S.J.T. Optic elements including lenses, molds, and 3-D printed components were designed by S.J.T. The project was designed and directed by J.M.C in close collaboration with S.J.T., H.D.P., M.R.J., Z.J. and M.L.C. The paper was written by J.M.C. with input and revisions from all authors.

## Competing interests

S.J.T., H.D.P, and J.M.C declare competing financial interests as inventors on patent applications covering the synthesis and processing of polysulfides for thermal imaging applications (AU2022900289, AU2024902135, and AU2025900891). No other authors declare a competing interest.
