## [Transparent Peer Review file · Nature Communications]

Thermal Imaging Using Sulfur Polymer Optics

Corresponding Author: Professor Justin M. Chalker

Version 0:

Reviewer comments:

Reviewer #1

(Remarks to the Author)

Reviewer #2

(Remarks to the Author)

This article demonstrated a route to fabricate sulfur polymer optics with higher MWIR and LWIR transparencies than previous reported sulfur polymers. This work identified the formation of cyclopropane rearrangement products instead of the expected cyclopropane microstructures as reported. Using monomers with pre-installed C-S bonds, expected cyclopropane microstructures were obtained by reacting the monomers with elemental sulfur. Sulfur polymer optics with advanced transparencies in MWIR and LWIR regions were fabricated. The article is suggested to be accepted after addressing the below comments:

1. What is the sulfur content of the sulfur polymers used in Fig. 2e, 2f and 2g? The sulfur content is vital for affecting the IR transparencies.
2. What is the rule for adjusting the T_g of the sulfur polymers used for optics? It is mentioned that the T_g should be above 100 °C to facilitate the processing of the optics. How to balance the high T_g while maintaining the good transparencies in MWIR and LWIR region?

Reviewer #3

(Remarks to the Author)

The manuscript describes the synthesis, properties, and application of sulfur-norbornene copolymer, a material which, while previously proposed computationally, has been synthetically inaccessible through the reaction between norbornadiene and elemental sulfur. The authors provide an elegant solution to this issue through first isolating the cyclic precursors (trisulfide and pentasulfide monomers) and subjecting these compounds to thermal copolymerization with elemental sulfur. The resulting polymers exhibit high glass transition temperatures even at high sulfur contents. Additionally, the polymers exhibit good LWIR transparency and could be shaped into windows and optical lenses. I believe this work provides a significant advancement in the area, not only in terms of providing polymeric materials with a combination of good thermal and optical properties in the MWIR and LWIR region, but also in terms of demonstrating a distinct way of producing high sulfur content polymers. The experimental results provided in the manuscript support the claims, and there is enough detail for the work to be reproduced. I do, however, have a few points of concern listed below.

1. It has been previously reported that some high sulfur content polymers lack long-term stability. Significant mass loss and fluctuations in glass transition temperatures have been reported for polymers prepared from sulfur and diisopropenyl benzene (See ACS Macro Lett 2019, 8, 1670). With the sulfur content of the polymers described by the authors being higher, long-term stability (in terms of mass loss, sulfur content, T_g changes, refractive index changes, and etc) should be provided.
2. The authors state in the abstract that “much higher MWIR and LWIR transparencies are required to compete with industry standards.” A major advantage of inorganic lenses is that they can be shaped through mechanical polishing and grinding. This is the key for producing precision optical lenses which can be integrated into multi-lens assemblies in camera modules.

The authors mention that sulfur-norbornadiene polymer samples were polished before optical measurements were made, and this was done using sandpaper. Additional comments about the applicability of this new polymer to conventional precision lens shaping processes should be discussed.

3. Figure S69: The y axis label reads “transmittance (%)” but the manuscript suggests that this should be fractional transmittance, not percent transmittance.

Version 1:

Reviewer comments:

Reviewer #1

(Remarks to the Author)

I have reviewed the authors' responses to my previous comments and find them satisfactory. No further revisions or clarifications are required. I recommend acceptance of the manuscript.

Reviewer #2

(Remarks to the Author)

This article presents a significant and valuable contribution to the field of materials science for thermal imaging applications, and I strongly recommend its acceptance for publication.

We thank the Reviewers for their detailed and highly constructive feedback. We have carried out additional experiments based on these suggestions and provide a point-by-point response to each comment and question. Responses are provided in blue text. Additionally, we have provided a combined and annotated manuscript and supplementary information PDF file in which all changes are highlighted in yellow.

Reviewer 1

Recommendation: Reconsider after major revisions.

I have carefully reviewed the manuscript entitled "Thermal Imaging Using Sulfur Polymer Optics". The manuscript is well-written, methodologically sound, and addresses an important problem in infrared optics the high cost and scarcity of conventional materials like Ge and ZnSe. The authors' approach, centered on the pre-installation of C–S bonds to avoid cyclopropane-related absorption, is chemically elegant and convincingly executed. The work is interesting and it is proposed for publication after major revisions and necessary explanations.

Thank you for this positive assessment and summary of our study.

1. While the reported LWIR transmission is the highest to date for similar sulfur polymers, absolute transmission values (e.g., ~19% at 1 mm thickness) are still modest. Could the authors elaborate on possible strategies such as AR coatings or compositional tuning to further improve transmittance?

This is a valid and important point. Antireflective (AR) coatings are routinely used on germanium, silicon and chalcogenide glass lenses. These coatings reduce reflection and, as a consequence, ensure more infrared light passes through the lens to the camera detector. For sulfur polymer lenses, some light is indeed reflected, and application of an AR coating of a precise thickness and refractive index could further improve transmission. For this study, our focus was to make lenses of the simplest composition with the aim to provide a low-cost optic suitable for consumer-based sensors and cameras used in products such as smart appliances, air conditioning units, or fire detectors. In these cases, the lens does not necessarily need to have an AR coating if the polymer lens alone is sufficient for thermal detection or rudimentary imaging. Furthermore, we aimed to make a polymer lens that could be molded in a single step with high throughput—another advantage for sensors or cameras that are mass produced for consumer products. We think that the lenses we report meet this goal in this initial study.

With that said, our next goal is to push the limits of polymer optics in infrared imaging. For high-end applications such as firefighting, defense operations, high-resolution security cameras, and planetary imaging and space applications, far greater transmission is required—rightfully noted by Reviewer 1. We plan to take two approaches in our next project in an attempt meet this goal. First, we will make antireflective coatings using a thin layer or layers of *the same polymer used to make the lens* (or perhaps a different sulfur copolymer). The refractive index can be controlled by variation of the sulfur content, which is one of the enabling features of these polymers. Furthermore, through S-S exchange reactions, this polymer layer could be *chemically bonded* to the polymer lens. In this way, the AR coating is not really a coating, but a boundary layer of the monolithic polymer lens with a different refractive index. Gradients of refractive index at the surface of the lens should also be possible, which is an exciting prospect in contrast to multiple layers of different AR materials. In the conclusion of the manuscript, we discuss this approach as an area for future work.

Another approach to improve LWIR transmission is to change the structure of the organic domain of the polymer. The norbornane microstructure was specifically selected for this project because of Pyun's computational prediction that it should have excellent LWIR transmittance. We also wanted to address the synthetic challenge to make this polymer, which has so far not been possible due to the complex rearrangements encountered in the reaction of norbornadiene and sulfur. We are pleased to report our strategy to make this polymer and validate Pyun's prediction of its utility in IR imaging in this study. In future work, we will explore other organic microstructures that have far lower absorbance in the 7-14 μm range. Our strategy is to design monomers that have few or no C-H bonds, no alkenes, and no other functional groups that absorb LWIR light. Additionally, this organic component of the polymer should have the lowest molecular weight possible, so the mass required when polymerizing with sulfur is also as low as possible. This strategy should approach the maximum LWIR transmission for a sulfur copolymer. The challenge is to also have this organic unit in the polymer maintain the high T_g required to maintain the precise shape of the lens and its mechanical and thermal stability. We will report our progress in this next phase in due course. In the conclusion of our manuscript, we have added a comment on this goal for future studies.

We also note that in some situations thermal stability of the lens may be a priority over LWIR transmission, such as monitoring of extreme environments such as wildfires, furnaces, nuclear reactors and missions to Venus. We are not claiming that our lenses would withstand these environments in their current form, but this is a case where the AR coating may be less important than thermal stability. So greater thermal stability is also an area for future research. For this study, we just focused on making the polymer lenses and validating their use in thermal imaging using the FLIR Lepton development module.

2. While qualitative LWIR images were demonstrated, are there any quantitative metrics (e.g., modulation transfer function, SNR, thermal sensitivity) available for comparison with silicon or germanium optics?

Thank you for this important suggestion. To quantitatively compare our polymer **1** lens to the silicon doublet in the FLIR Lepton 3.5, we carried out noise equivalent thermal difference (NETD) measurements and relative illumination experiments. The NETD and relative illumination experiments can be found in the supplementary information on pages S105-S109. These new data are also discussed in the main text (page 7-8).

The NETD is a measure of the sensitivity of a thermal camera with a lower value indicating that smaller temperature differences can be detected. A standardized method was developed to determine the NETD for several polymer **1** lenses and the commercial silicon doublet provided with the FLIR Lepton. The FLIR Lepton 3.5's silicon doublet lens has a focal length of 1.77 mm and an f-number of 1.1. It was found that the polymer **1** lenses with a low f-number had a similar NETD to the silicon lens (polymer lenses 2 and 4 were within 10 mK of the silicon doublet). The polymer **1** lenses also displayed the expected behavior that their NETD values were proportional to the square of the f-number. The methods and data in this comparison are provided in the supplementary information on pages S105-S106.

The other metric that we used to compare with the silicon lens was relative illumination. This is a valuable property of a lens which indicates how the intensity of light changes from the center to the edge of an image. All the polymer **1** lenses had a simple plano convex design. As expected, the wide field of view lenses had a significant decrease in relative illumination near the edge of the image due to field curvature. However, the lenses with a longer focal length and narrower field of view performed much closer to that of the silicon lens. These results indicate

that a lens design which accounts for field curvature could likely perform similarly to silicon in relative illumination. While we understand the difficulties in designing a lens with both a low f-number and high relative illumination performance, these results give us confidence that future lens designs will approach commercially available lenses. We also note that freeform molding is a viable pathway to address illumination correction. Relative illumination testing and data are provided in the supplementary information on pages S107-S109.

3. Have any investigations been conducted regarding the compatibility of Polymer 1 with conventional antireflective (AR) coating materials. If not, it would be helpful for the authors to discuss potential challenges related to surface energy or adhesion properties that may need to be addressed to enable effective coating integration.

The issue of coating adhesion is a potential challenge for applying conventional AR coatings to our polymer lenses. For this reason, we plan to take a non-conventional approach—as discussed in our response to question 1. Specifically, we will make antireflective coatings using a thin layer or layers of the same polymer used to make the lens (or perhaps a different sulfur copolymer). The refractive index can be controlled by variation of the sulfur content, which is one of the enabling features of these polymers. Furthermore, through S-S exchange reactions, this polymer layer could be chemically bonded to the polymer lens. In this way, the AR coating is not really a coating, but a boundary layer of the monolithic polymer lens with a different refractive index. This would also prevent peeling or loss of adhesion that can occur with traditional coatings. Gradients of refractive index at the surface of the lens should be possible, which is an exciting prospect. In the conclusion of the manuscript, we discuss this approach as an area for future work.

4. A high and stable refractive index ($n \approx 1.87$) is valuable. Has the refractive index been shown to be consistent across batches or sample geometries? Were there any variations in index uniformity observed during lens fabrication or imaging tests?

This is an important question, as stability and reproducibility are critically important for polymer optics. We have therefore assessed the long-term stability of polymer 1 and lenses made from polymer 1. There were negligible differences in the T_g , TGA profile, and refractive index of freshly made polymer 1 and polymers made 6 or more months prior. Furthermore, a lens made 12 months ago was re-tested in the FLIR Lepton 3.5 and the images obtained were the same. These results suggest that polymer 1 is stable when stored at room temperature, and this composition does not suffer from any deformations, degradation, or sulfur blooming that would otherwise compromise image quality. Regarding reproducibility, six independently synthesized batches of polymer 1 all had the same refractive indices. These additional experiments are provided in the supplementary information on pages S114-S117 and discussed in the main text (page 8). We also note that the LWIR refractive index of 1.87 involves smaller chromatic aberration corrections compared to Ge (~ 4.0) or ZnSe (2.6) and can simplify lens system design in simple optical systems.

Reviewer 2

This article demonstrated a route to fabricate sulfur polymer optics with higher MWIR and LWIR transparencies than previous reported sulfur polymers. This work identified the formation of cyclopropane rearrangement products instead of the expected cyclopropane microstructures as reported. Using monomers with pre-installed C-S bonds, expected cyclopropane microstructures were obtained by reacting the monomers with elemental sulfur. Sulfur polymer optics with advanced transparencies in MWIR and LWIR regions were fabricated. The article is suggested to be accepted after addressing the below comments:

Thank you for this positive assessment of our study.

1. What is the sulfur content of the sulfur polymers used in Fig. 2e, 2f and 2g? The sulfur content is vital for affecting the IR transparencies.

Thank you for noticing this important point. All of the characterization data in Figure 2 is from the composition of polymer **1** containing 81% sulfur by mass. This composition was specified in the main text when referring to Figure 2, but this composition was not specified in the actual figure or caption. We have revised the caption to Figure 2 to make it clear that all data in this figure were obtained from the polymer containing 81% sulfur.

2. What is the rule for adjusting the T_g of the sulfur polymers used for optics? It is mentioned that the T_g should be above 100 °C to facilitate the processing of the optics. How to balance the high T_g while maintaining the good transparencies in MWIR and LWIR region?

This is an important question, of general relevance to the design of copolymers made from sulfur. There are no simple rules for all such polymers, but for polymer **1** the following trends were determined after the synthesis and analysis of polymers with varying sulfur content:

Glass transition temperature: The T_g for polymer **1** decreases with increasing sulfur content. For example, if monomer **9** is polymerized directly without any added elemental sulfur, the T_g is very high at 181 °C. This polymer has a sulfur content of 67% by mass (the same as monomer **9**). With increasing sulfur in polymer **1**, the T_g decreases: 72% sulfur has a $T_g = 164$ °C and a polymer composition of 81% sulfur has a $T_g = 154$ °C (measured by DMA) and $T_g = 115$ °C (measured by DSC with slow 10 °C/min ramp rate). In general, all of these polymers have T_g values > 100 °C, with higher sulfur content correlated with a reduced T_g (See supplementary information pages S64-S65 for additional details).

MWIR and LWIR transmission: The norbornane microstructure in polymer **1** is responsible for the majority of MWIR and LWIR absorbance. These absorbances are due to the modes of vibration associated with C-H and C-C bonds in the fingerprint region of the IR spectrum (see Figure 2e and Figure S69, for instance). Therefore, reducing the amount of the norbornane microstructure in the polymer will increase MWIR and LWIR transmission.

There is therefore a balance between having enough norbornane in polymer **1** to have a T_g suitable for the intended application, while also maintaining the required MWIR and LWIR transmission for generating a quality image. In this project, a $T_g > 100$ °C was targeted for the shape persistence required for making and evaluating stable lenses. At this composition, the sulfur content is still relatively high (81% by mass), so LWIR transmission is suitable for imaging using the polymer as the only optic in the thermal imaging camera.

Two comments on the composition and the trade-off between T_g and LWIR transmission for polymer **1** were added to the main text (page 4).

Reviewer 3

The manuscript describes the synthesis, properties, and application of sulfur-norbornene copolymer, a material which, while previously proposed computationally, has been synthetically inaccessible through the reaction between norbornadiene and elemental sulfur. The authors provide an elegant solution to this issue through first isolating the cyclic precursors (trisulfide and pentasulfide monomers) and subjecting these compounds to thermal copolymerization with elemental sulfur. The resulting polymers exhibit high glass transition temperatures even at high sulfur contents. Additionally, the polymers exhibit good LWIR transparency and could be shaped into windows and optical lenses. I believe this work provides a significant advancement in the area, not only in terms of providing polymeric materials with a combination of good thermal and optical properties in the MWIR and LWIR region, but also in terms of demonstrating a distinct way of producing high sulfur content polymers. The experimental results provided in the manuscript support the claims, and there is enough detail for the work to be reproduced. I do, however, have a few points of concern listed below.

Thank you for this detailed and supportive assessment of our study.

1. It has been previously reported that some high sulfur content polymers lack long-term stability. Significant mass loss and fluctuations in glass transition temperatures have been reported for polymers prepared from sulfur and diisopropenyl benzene (See ACS Macro Lett 2019, 8, 1670). With the sulfur content of the polymers described by the authors being higher, long-term stability (in terms of mass loss, sulfur content, T_g changes, refractive index changes, and etc) should be provided.

This is an excellent point. If polymer 1 lenses deformed, suffered from sulfur loss or blooming, or changed in refractive index, the LWIR imaging quality would also deteriorate. We have therefore re-tested the polymer and lenses after storing for 6-12 months. Fortunately, there were negligible differences in the T_g, TGA profile, and refractive index of freshly made polymer 1 and polymers made 6 or more months prior. Furthermore, a lens made 12 months ago was re-tested in the FLIR Lepton 3.5 and the images obtained were the same. These results suggest that polymer 1 is stable when stored at room temperature, and this composition does not suffer from any deformations, degradation, or sulfur loss or blooming that would otherwise compromise image quality. These additional experiments are provided in the supplementary information on pages S114-S117 and discussed in the main text (page 8).

We attribute the stability of polymer 1 to reaction conditions that ensure complete reaction of elemental sulfur and the high glass transition temperature of polymer 1. With a high T_g, the mobility of the polysulfide chains will be much less than many of the polymers made from inverse vulcanization that have a lower T_g. We suspect that the lower chain mobility due to high T_g prevents S-S cleavage reactions or sulfur blooming at room temperature. We also note that even though the optimal composition is a very high 81% sulfur by mass, the average sulfur rank is only ~6 based on the stoichiometry for the monomer and sulfur (this is a consequence of the low molar mass of the norbornane unit in the polymer microstructure).

In the reference noted by Reviewer 3, a similar observation was reported in which a high T_g polymer made by inverse vulcanization exhibited greater thermal stability. For this reason, we have added the following statement in the main text on page 8, and cite this important previous study (reference 34):

“The long-term stability of polymer 1 and polymer 1 optics is critical for consistent production of high-quality thermal images. We therefore measured the T_g, TGA profile, and refractive index for samples of polymer after 6 or more months of storage. Negligible differences were

observed between these samples and freshly prepared samples of polymer **1** (Supplementary Figs. S109-S111). Additionally, a 12-month-old polymer lens was re-tested in the FLIR Lepton 3.5 module, and the image quality was the same as when first tested 1 year prior (Supplementary Fig. S112). We attribute the stability of **1** to its high T_g , which is consistent with studies of inverse vulcanization that suggest high T_g is correlated with greater thermal stability and reduced sulfur loss during processing and storage³⁴.”

2. The authors state in the abstract that “much higher MWIR and LWIR transparencies are required to compete with industry standards.” A major advantage of inorganic lenses is that they can be shaped through mechanical polishing and grinding. This is the key for producing precision optical lenses which can be integrated into multi-lens assemblies in camera modules. The authors mention that sulfur-norbornadiene polymer samples were polished before optical measurements were made, and this was done using sandpaper. Additional comments about the applicability of this new polymer to conventional precision lens shaping processes should be discussed.

Grinding, polishing and milling are indeed commonly used in the manufacture of lenses made from inorganic materials, including those materials used in thermal imaging optics. This is particularly important for complex optics such as aspheric lenses. Our polymer lens can also be polished by such methods, as demonstrated on the windows used for refractive index measurements (page S79 and S116) and infrared transmission testing (pages S73-S77), and also on the lenses made by reactive compression molding (page S111-S112). Both sandpaper (with progressively finer grit) and a micro gloss polish with 1-micron abrasive crystals were used to achieve optical quality finishes. Because of the glassy properties and high T_g , we think that precision milling using diamond point turning should also be possible, and this is specifically the reason we made the large preforms shown in Figure 2i. We were not able to test this directly as we do not have access to such equipment at this time. Commenting on this issue we have added the following in the main text:

Page 6: “With ample amounts of monomers **9** and **10** in hand, polymer **1** was fashioned into freestanding windows and preforms by casting the polymerization mixture into suitable molds and curing. Due to the high glass transition temperature of **1**, it could be polished to provide optical quality surfaces using a micromesh polishing kit and micro gloss polish (Supplementary Fig. S69 and page S79).”

Conclusion (page 10): “Future work will focus on upscaling the synthesis of **1** and making more complex optics, to deliver increased image quality required for higher-end applications in navigation, defense, security systems, space applications, firefighting, and high-temperature monitoring in furnaces and nuclear reactors. For preparing more complex prototype lenses, the high T_g and stability of **1** is anticipated to allow construction of more complex architectures by diamond point turning of polymer preforms.”

More generally, we think one advantage of polymer **1** over traditional inorganic lenses is that it can be processed by molding techniques used in the mass production of plastic products. These molding techniques have higher throughput than subtractive milling processes. We demonstrate this in Figure 3c where we mold 89 lenses in a 1-minute molding process. Higher throughput would be beneficial for applications that require low-cost and high-volume manufacture, for instance in consumer products that use LWIR sensors or cameras in smart appliances. We have clarified the benefit of high-throughput molding for certain cases on page 8 in the main text:

“The ability to mold pre-made polymer **1** bodes well for high-throughput lens manufacture, which could be an advantage in some scenarios in comparison to slower, subtractive milling processes typically used for inorganic optics.”

3. Figure S69: The y axis label reads “transmittance (%)” but the manuscript suggests that this should be fractional transmittance, not percent transmittance.

Thank you for identifying this error. The label for the y-axis in Figure S69 has been changed to “fractional transmittance”.

Thermal Imaging Using Sulfur Polymer Optics

Recommendation: Reconsider after major revisions.

Comment

I have carefully reviewed the manuscript entitled "Thermal Imaging Using Sulfur Polymer Optics". The manuscript is well-written, methodologically sound, and addresses an important problem in infrared optics the high cost and scarcity of conventional materials like Ge and ZnSe. The authors' approach, centered on the pre-installation of C-S bonds to avoid cyclopropane-related absorption, is chemically elegant and convincingly executed. The work is interesting and it is proposed for publication after major revisions and necessary explanations.

A list of necessary changes is given bellow:

1. While the reported LWIR transmission is the highest to date for similar sulfur polymers, absolute transmission values (e.g., ~19% at 1 mm thickness) are still modest. Could the authors elaborate on possible strategies such as AR coatings or compositional tuning to further improve transmittance?
2. While qualitative LWIR images were demonstrated, are there any quantitative metrics (e.g., modulation transfer function, SNR, thermal sensitivity) available for comparison with silicon or germanium optics?
3. Have any investigations been conducted regarding the compatibility of Polymer 1 with conventional antireflective (AR) coating materials. If not, it would be helpful for the authors to discuss potential challenges related to surface energy or adhesion properties that may need to be addressed to enable effective coating integration.
4. A high and stable refractive index ($n \approx 1.87$) is valuable. Has the refractive index been shown to be consistent across batches or sample geometries? Were there any variations in index uniformity observed during lens fabrication or imaging tests?

This article demonstrated a route to fabricate sulfur polymer optics with higher MWIR and LWIR transparencies than previous reported sulfur polymers. This work identified the formation of cyclopropane rearrangement products instead of the expected cyclopropane microstructures as reported. Using monomers with pre-installed C-S bonds, expected cyclopropane microstructures were obtained by reacting the monomers with elemental sulfur. Sulfur polymer optics with advanced transparencies in MWIR and LWIR regions were fabricated. The article is suggested to be accepted after addressing the below comments:

1. What is the sulfur content of the sulfur polymers used in Fig. 2e, 2f and 2g? The sulfur content is vital for affecting the IR transparencies.
2. What is the rule for adjusting the T_g of the sulfur polymers used for optics? It is mentioned that the T_g should be above 100 °C to facilitate the processing of the optics. How to balance the high T_g while maintaining the good transparencies in MWIR and LWIR region?

This article presents a significant and valuable contribution to the field of materials science for thermal imaging applications, and I strongly recommend its acceptance for publication.